# Anti-Inflammatory, Anti-Oxidative and Anti-Apoptotic Effects of Thymol and 24-Epibrassinolide in Zebrafish Larvae

**DOI:** 10.3390/antiox12061297

**Published:** 2023-06-18

**Authors:** Germano A. B. Lanzarin, Luís M. Félix, Sandra M. Monteiro, Jorge M. Ferreira, Paula A. Oliveira, Carlos Venâncio

**Affiliations:** 1Centre for the Research and Technology of Agro-Environment and Biological Sciences (CITAB), University of Trás-os-Montes and Alto Douro (UTAD), 5000-801 Vila Real, Portugal; glanzarin@utad.pt (G.A.B.L.); smonteir@utad.pt (S.M.M.); pamo@utad.pt (P.A.O.); 2Inov4Agro, Institute for Innovation, Capacity Building and Sustainability of Agri-Food Production, University of Trás-os Montes and Alto Douro (UTAD), 5000-801 Vila Real, Portugal; 3Department of Biology and Environment, School of Life and Environmental Sciences, University of Trás-os Montes and Alto Douro (UTAD), 5000-801 Vila Real, Portugal; 4Instituto de Investigação e Inovação em Saúde (i3s), Laboratory Animal Science (LAS), Instituto de Biologia Molecular Celular (IBMC), University of Porto (UP), 4200-135 Porto, Portugal; jorge.ferreira@i3s.up.pt; 5Department of Veterinary Sciences, School of Agrarian and Veterinary Sciences, University of Trás-os-Montes and Alto Douro (UTAD), 5000-801 Vila Real, Portugal; 6Department of Animal Science, School of Agrarian and Veterinary Sciences, University of Trás-os-Montes and Alto Douro (UTAD), 5000-801 Vila Real, Portugal

**Keywords:** natural products, anti-inflammatory, antiapoptotic, oxidative stress, zebrafish

## Abstract

Thymol (THY) and 24-epibrassinolide (24-EPI) are two examples of plant-based products with promising therapeutic effects. In this study, we investigated the anti-inflammatory, antioxidant and anti-apoptotic effects of the THY and 24-EPI. We used zebrafish (*Danio rerio*) larvae transgenic line (Tg(*mpx*GFP)^i114^) to evaluate the recruitment of neutrophils as an inflammatory marker to the site of injury after tail fin amputation. In another experiment, wild-type AB larvae were exposed to a well known pro-inflammatory substance, copper (CuSO4), and then exposed for 4 h to THY, 24-EPI or diclofenac (DIC), a known anti-inflammatory drug. In this model, the antioxidant (levels of reactive oxygen species—ROS) and anti-apoptotic (cell death) effects were evaluated in vivo, as well as biochemical parameters such as the activity of antioxidant enzymes (superoxide dismutase, catalase and glutathione peroxidase), the biotransformation activity of glutathione-S-transferase, the levels of glutathione reduced and oxidated, lipid peroxidation, acetylcholinesterase activity, lactate dehydrogenase activity, and levels of nitric acid (NO). Both compounds decreased the recruitment of neutrophils in Tg(*mpx*GFP)^i114^, as well as showed in vivo antioxidant effects by reducing ROS production and anti-apoptotic effects in addition to a decrease in NO compared to CuSO4. The observed data substantiate the potential of the natural compounds THY and 24-EPI as anti-inflammatory and antioxidant agents in this species. These results support the need for further research to understand the molecular pathways involved, particularly their effect on NO.

## 1. Introduction

Natural plant products have been used for medicinal purposes since ancient times and present a challenge for the discovery of new drugs [1,2]. Plants produce large amounts of metabolites with therapeutic values, such as alkaloids, steroids, saponins, flavonoids, and many others [3]. Several plant compounds have been intensively studied in the search for anti-inflammatory properties with fewer side effects [4,5]. As is the case of thymol (THY) (2-isopropyl-5-methylphenol), a natural terpenoid that has been used in traditional medicine, whose polypharmacological properties such as anti-inflammatory, antioxidant, local anesthetic, healing, antinociceptive action, among others, have been earlier described [6,7]. Additionally, it has been used in several preventive therapies, such as neurological, cardiovascular, and rheumatological diseases [8]. However, recently pro-oxidant and apoptotic effects of THY were reported [6], which highlighted the need for further studies to clarify their therapeutic effects, safety and toxicity profile, and fully understand their mechanism of action [8]. Another compound that appears to be promising for therapeutic purposes is 24-epibrassinolide (24-EPI), a natural phytohormone found in plants [3,9]. In addition, 24-EPI belongs to the group of polyhydroxy steroid hormones called brassinosteroids, which have similarities to cholesterol-derived animal steroids [3,10]. Its biological properties include neuroprotective, antiviral, antioxidant, and anti-apoptotic effects [10,11,12,13,14,15]. However, its anti-inflammatory properties have not yet been elucidated, contrary to what has already been described for other brassinosteroids [3]. 

The inflammatory process is a reaction of the immune system in which neutrophils migrate to the affected site, performing activities such as degranulation, phagocytosis, reactive oxygen species (ROS) production, secretion of pro-inflammatory cytokines, and extrusion of extracellular neutrophil traps [16,17]. Inflammation results from complex dynamic changes in cellular, molecular, and physiological parameters. Therefore, they cannot be effectively simulated in vitro, requiring the use of in vivo models [18,19]. 

Zebrafish is a relevant animal model for immunological studies as it shares similar characteristics with species of interest, in particular with humans, having in common several biologic pathways, such as inflammation response [20]. Since zebrafish is a widely used animal model in pharmacological studies, it plays a key role in the evaluation of new drugs [21,22]. In particular, the transgenic zebrafish strain (Tg(*mpx*GFP)^i114^), where the neutrophils are labelled with the specific green fluorescent protein (GFP), allowing the efficient monitoring of neutrophil migration activity [17,23]. 

The chemical induction of inflammation in zebrafish larvae after exposure to copper (CuSO_4_) has been previously characterized and involves an oxidative process [24,25,26]. The present study aims to evaluate the anti-inflammatory, anti-oxidative and anti-apoptotic effects of the natural compounds THY and 24-EPI in two different strains of zebrafish larvae using CuSO4 as a chemical inflammation inducer, making comparisons with a known non-steroidal anti-inflammatory drug (NSAID), diclofenac (DIC). In this way, the Tg(*mpx*GFP)^i114^ larvae were used to evaluate the anti-inflammatory level of THY and 24-EPI through the study of neutrophil recruitment to the site of the tail lesion. In addition, we used the AB strain (Wild Type, AB-WT), exposed to natural compounds and in combination with CuSO_4_, to analyze several biochemical markers, such as oxidative stress and cell death. It was hypothesized that some prevention of the CuSO_4_-induced changes at the cellular level through exposure to the natural compounds.

## 2. Materials and Methods

### 2.1. Chemicals and Solutions

E3 medium (0.5 mM NaCl, 0.017 mM KCl, 0.033 mM CaCl_2_, 0.033 mM MgSO_4_, pH 7.0–7.4) was used to prepare all solutions. Copper sulphate pentahydrate (CuSO_4_.5H_2_O) (Merck, S.A., Algés, Portugal) was used as the source of copper and a stock solution of 4 mM (1000 mg/L) was prepared in ultra-pure water and further diluted in E3 medium to 10 µM (2.5 mg/L) [27]. Tricaine methanesulfonate (MS-222) (Sigma Aldrich, Lisboa, Portugal) stock solution of 5.74 mM (1500 mg/L) was prepared in buffered water, pH was adjusted to 7.4 and diluted in E3 medium to 574 µM (150 mg/L), according to [28,29]. Diclofenac (DIC) sodium salt (Alfa Aesar, Kandel, Germany) stock solution was prepared in E3 medium at 2 mM (636 mg/L) and diluted to 1.5 µM (4.75 mg/L) [26]. THY (5-methyl-2-propan-2-ylphenol) (Sigma Aldrich) stock solution was prepared in ultra-pure water containing 10% ethanol and diluted in E3 medium of 430 µM (64.5 mg/L) and further diluted to 20 µM (3 mg/L). A 20 µM (9.6 mg/L) stock solution of 24-EPI (Sigma Aldrich, Lisboa, Portugal) was prepared in ultra-pure water containing 5% ethanol and diluted to 5 µM (2.4 mg/L) in E3 medium. All other reagents used were purchased from Sigma Aldrich (Lisboa, Portugal) or of the highest commercially available grade.

### 2.2. Maintenance and Reproduction of Zebrafish

The maintenance and reproduction of zebrafish was carried out following the European Directive 2010/63/EU and Portuguese legislation (DL 133/2013) on animal welfare and the procedures involving manipulation of adults for reproduction were approved by National authority DGAV (Direção Geral de Alimentação e Veterinária) through the project license 014703/2017-06-16. Zebrafish fertilized eggs of the Transgenic (Tg(*mpx*GFP)^i114^) strain, showing specific neutrophil fluorescence, were obtained from European Zebrafish Resource Center, Karlsruhe Institute of Technology, Germany. Wild-Type AB (WT) and Transgenic (Tg(*mpx*GFP)^i114^) adult zebrafish (*Danio rerio*) were maintained in the premises of the University of Trás-os-Montes and Alto Douro, Vila Real. The water was maintained at 28 ± 0.5 °C, pH 7.5–8, dechlorinated, aerated, charcoal-filtered and UV-sterilized, and was provided by municipal water of the city of Vila Real. The zebrafish were subjected to a circadian cycle (14 h light/10 h dark) with the lights on at 8:00 am and were fed twice daily with a commercial diet (Zebrafeed^®^, Sparos Lda, Olhão, Portugal). The breeding activity was promoted by the union of zebrafish overnight with a ratio of 2 males to 1 female with the spawning induced by the morning light and the eggs collected after 1 h. Chloramine-T (0.5% *w*/*v*) [30] was used to disinfect the eggs and those with normal morphology were selected randomly and distributed in 6-well plates and maintained in E3 medium at 28.5 °C until they reached 72 hpf. The solution was renewed daily and larvae showing some type of abnormality or mortality were removed.

### 2.3. Experimental Design and Treatments

The experimental procedures of this study took place in three phases. In the first phase, the Tg(*mpx*GFP)^i114^ larvae were used, while in the second and third phases the WT larvae were used. In all phases, the solution used for the control group was E3 medium. In the first phase (Figure 1A), the tail section of the larvae was cut, and larvae were exposed to the different compounds under analysis to assess the neutrophil migration response. As a pro-inflammatory agent, 10 µM CuSO_4_ was used, as already described in [26,31]. As negative control, the anti-inflammatory agent diclofenac (1.5 µM DIC) was used according to the concentration previously described [26]. The concentrations chosen for the study of the THY and 24-EPI (20 and 5 µM, respectively) were selected according to pilot studies whose results are presented in Appendix A and also based on a THY concentration that has a safer profile on larval development [6]. For the second (Figure 2A) and third (Figure 3A) phases, the same experimental analysis was performed, in which the WT larvae were exposed to 10 μM CuSO_4_ for 30 min, to induce a systemic inflammatory process [25,26]. Subsequently, the larvae were exposed to the pharmacological treatments described in the first phase (DIC, THY and 24 EPI), in order to evaluate, in a second phase, the levels of ROS and apoptosis, and in a third phase, the biochemical markers. The larvae were thus divided into eight groups: E3; 10 µM CuSO_4_; 1.5 µM DIC; CuSO_4_ + DIC; 20 µM THY; CuSO_4_ + THY; 5 µM 24-EPI; CuSO_4_ + 24-EPI. The evaluations were carried out after 4 h of exposure, in 6-well plates, a period after which it is described that CuSO_4_ induces an initial inflammatory process [24,25,31].

### 2.4. Tail Transection and Neutrophil Migration Count 

As described in the experimental design, this first phase of the study was carried out following methods previously described with some modifications [32,33,34]. Tg(*mpx*GFP)^i114^ larvae (72 hpf) were exposed to each compound separately according to their respective groups (E3; 10 µM CuSO_4_, 1.5 µM DIC, 20 µM THY and 5 µM 24-EPI) for 30 min, preceding tail transection. At least five replicates of five larvae were used per group. Then, the larvae were anesthetized by immersion in 574 µM of MS-222, and the complete transection of the tail was performed with a sterile scalpel. Posteriorly, the larvae were washed and replaced in each solution containing their respective compounds and exposed for 4 h post-injury (hpi). Then, fluorescent images were obtained under an inverted microscope (IX 51, Olympus, Antwerp, Belgium) equipped with an Olympus U-RFL-T fluorescent light source (Olympus, Antwerp, Belgium) and FITC filter, using a 4X Olympus UIS-2 objective lens (Olympus Co., Ltd., Tokyo, Japan). The data were acquired using the Cell R software (Olympus, Antwerp, Belgium) and the fluorescent images were processed with Adobe Photoshop CS6 (Adobe Systems, San Jose, CA, USA). Neutrophil migration was quantified in an area of 250 µm from the cut using the ImageJ2 program (version 2.0.0, National Institutes of Health of the USA, Bethesda, MD, USA) and through an automatic cell count extension (Find Maxima) [35].

### 2.5. ROS and Apoptosis Analysis in AB-WT Larvae

In the second phase of the study, to detect ROS and programmed cell death, AB-WT larvae with 72 hpf were exposed to 10 µM CuSO_4_ for 30 min. Thereafter, the larvae were divided into the eight treated groups with different combinations of the pharmacological compounds (E3; 10 µM CuSO_4_; 1.5 µM DIC; CuSO_4_ + DIC; 20 µM THY; CuSO_4_ + THY; 5 µM 24-EPI; CuSO_4_ + 24-EPI). In addition, 4 h post-exposure (hpe), at least five replicates of 25 larvae were incubated in the dark with DCFH-DA (20 mg/L) or with AO (5 mg/L) solutions for 30 min at 28 °C, respectively, as described before [31,36,37]. After washing three times with E3 medium, some larvae were separated, examined and illustrative fluorescence images were captured, as described in the first phase. The remaining larvae were collected into tubes containing buffer (0.32 mM de sucrose, 20 mM de HEPES, 1 mM de MgCl_2_, and 0.5 mM de phenylmethyl sulfonylfluoride, pH = 7,4) at −80 °C. The samples were homogenized using the TissueLyzer II apparatus (Qiagen) followed by centrifugation for 20 min at 15,000× *g* (12.517 rpm) at 4 °C, then the supernatant was reused and transferred to clean tubes. Fluorescence intensity was measured using Varian Cary Eclipse (Varian, Palo Alto, CA, USA) Spectrofluorometer, equipped with a microplate reader, at excitation and emission wavelengths of 488/522 nm (ROS) and 488/515 nm (AO), respectively. Levels of induced ROS and apoptosis were expressed as a percentage of the control.

### 2.6. Study of Biochemical Markers in WT Larva

In the third phase of the study, WT larvae with 72 hpf were exposed to 10 µM CuSO_4_ during 30 min. Thereafter, the larvae were divided into the eight treated groups with different combinations of the pharmacological compounds (E3; 10 µM CuSO_4_; 1.5 µM DIC; CuSO_4_ + DIC; 20 µM THY; CuSO_4_ + THY; 5 µM 24-EPI; CuSO_4_ + 24-EPI). At 4 hpe the larvae were washed and frozen in buffer previously described in the second phase. For the determination of biochemical markers, at least five replicates of 50 larvae were collected and samples were prepared as previously reported [38,39,40]. Briefly, the samples were homogenized, the supernatant was then collected, and protein was measured at 280 nm using a Take3 Multi-Volume plate (Take3 plate, BioTek Instruments, Winooski, VT, USA). The samples (20 µL) were analyzed in duplicate at 30 °C in a PowerWave XS2 microplate scanning spectrophotometer (Bio-Tek Instruments, Winooski, VT, USA) or Varian Cary Eclipse (Varian, Palo Alto, CA, USA) Spectrofluorometer, equipped with a microplate reader. The determination of superoxide dismutase (SOD) activity was measured according to [41] by inhibiting the photochemical reduction in nitrobluetetrazolium at 560 nm, starting the reaction by adding xanthine oxidase. SOD of bovine erythrocyte was used as a basis for building a standard curve (0–5 U/mL). The activity of catalase (CAT) was determined according to [42] at 240 nm, being normalized using a standard curve of bovine catalase (0–3 U/mL). Glutathione S-Transferase (GST) activity was determined in 340 nm. Mixing 2,4-dinitrochlorobenzene (CDNB) with reduced glutathione (GSH) [43]. The activity of glutathione peroxidase (GPx) was determined at 340 nm [44]. Glutathione levels were measured fluorometrically through reduced (GSH) and oxidized (GSSG) glutathione states based on derivatization with ortho-phthalaldehyde at an excitation wavelength of 320 nm and 420 nm emission [45]. Results were estimated with a standard curve GSH and GSSG (0–100 µM). According to the ratio of the coefficient between GSH and GSSG, the oxidative stress index (OSI) was then calculated. Lipid peroxidation (LPO) was measured according to [46] by thiobarbituric acid reactive substances (TBARS) at 530 nm and 600 nm (non-specific). Oxidative phospholipid, malondialdehyde (MDA), was determined with a curve standard (0–100 µM) of malondialdehyde bis (dimethyl acetal). Acetylcholinesterase (AChE) was determined at 405 nm, according to [47]. Lactate dehydrogenase (LDH) was determined at 340 nm [48]. The level of nitric oxide (NO) was determined using the Griess method, as described by [49,50], with some modifications. Briefly, the samples were mixed with the Griess reagent in a 1:1 ratio and incubated for 15 min at room temperature, after which the absorbance was read at 546 nm. Sodium nitrate was used as the basis for the construction of a standard curve (0–1 µM).

### 2.7. Statistics

The data used were normalized to the values of the control group. The normality (Shapiro–Wilk) and variance homogeneity (Levene’s test) tests were used before differences between groups being analyzed using a unilateral analysis of variance (ANOVA) were followed by Tukey’s multiple comparison test. When the normality assumptions were not met, non-parametric Kruskal–Wallis analysis of variance was used, followed by the Dunn test with a Bonferroni adjustment for multiple comparisons. The IBM SPSS statistics version 26 for Windows was used to conduct the statistical tests and the differences were defined with *p* < 0.05.

## 3. Results

### 3.1. Neutrophils Migration

To assess the effects of inflammation of THY and 24-EPI, the number of neutrophils that migrated to the injury site was evaluated after 4 hpi of cutting the caudal fin in Tg(*mpxGFP*)^i114^ larvae with exposure to the compounds. Figure 1 showed that the four compounds affected the neutrophils migration in relation to the E3 (*F*(4,23) = 88.299 *p* = 0.0001). THY and 24-EPI induced a decrease in the number of neutrophils of 72% (*p* = 0.0001) and 40% (*p* = 0.0001), respectively, relative to the E3. The CuSO_4_, positive control (pro-inflammatory), induced an increase in the mean number of neutrophils by 35% (*p* = 0.0001), while treatment with DIC, negative control (anti-inflammatory), inhibited the recruitment of neutrophils to the lesion site by 55% (*p* = 0.0001), compared to the E3.

### 3.2. ROS Generation and Apoptosis In Vivo

Detection of ROS generation and cell death in 72 hpf AB-WT larvae exposed to CuSO_4_ for 30 min and then exposed for 4 hpe to the treatments was observed in vivo by DCFH-DA and AO staining, as shown in Figure 2. Regarding the ROS levels exposure of larvae to CuSO_4_ (*p* = 0.044), CuSO_4_ + DIC (*p* = 0.001), THY (*p* = 0.033) and CuSO_4_ + THY (*p* = 0.040) showed a decrease in their levels compared to the E3 (F(7,30 = 7.766 *p* = 0.0001). Regarding apoptosis, several alterations were observed in CuSO_4_ and E3 groups (F(7,30) = 25.814 *p* = 0.0001). Exposure to CuSO4 resulted in an increase in cell apoptosis compared to E3 group (*p* = 0.001). The groups that showed a decrease in the expression of apoptotic cells compared to E3 were THY (*p* = 0.0001), CuSO_4_ + THY (*p* = 0.009) and CuSO_4_ + 24-EPI (*p* = 0.0007). The groups that showed a decrease in the expression of apoptotic cells in relation to the CuSO4 were DIC (*p* = 0.005), CuSO4 + DIC (*p* = 0.0001), THY (*p* = 0.0001), CuSO4 + THY (*p* = 0.0001), 24-EPI (*p* = 0.0001) and CuSO4 + 24-EPI (*p* = 0.0001).

### 3.3. Study of Biochemical Markers

The evaluation of parameters related to oxidative stress in 72 hpf AB-WT larvae exposed to CuSO_4_ for 30 min and then exposed for 4 hpe to the treatments are shown in Figure 3 or in Appendix A. For all biochemical markers determined no changes were observed between the THY and 24-EPI groups in relation to the control group. Relative to SOD enzyme activity (X^2^(7,38) = 20.751, *p* = 0.04), the DIC (*p* = 0.035) and the CuSO_4_ + 24-EPI (*p* = 0.01) groups presented higher activity than the control group (E3). An increase in the activity of this enzyme was also shown in DIC (*p* = 0.001), CuSO_4_ + DIC (*p* = 0.022), 24-EPI (*p* = 0.004) and CuSO_4_ + 24-EPI (*p* = 0.0001) when compared to the CuSO_4_ group. Regarding the CAT enzyme activity (F(7,38) = 3.495, *p* = 0.005), a higher activity was observed in DIC (*p* = 0.01) and 24-EPI (*p* = 0.008) when compared to CuSO_4_ group. As for the biotransformation enzyme GST, none of the treatment groups showed significant differences (F(7,38) = 1.122, *p* = 0.370). The GPx activity showed only an increase in CuSO_4_ + THY compared to DIC (*p* = 0.037) (F(7,38) = 3.099, *p* = 0.011). No differences in glutathione levels were observed, GSH (F(7,38) = 1.610, *p* = 0.162) and GSSG (F(7,38) = 2.022, *p* = 0.077), a change was only observed in its OSI ratio, where the index of THY was lower than DIC (*p* = 0.04) (F(7.38) = 2.190, *p* = 0.57). No differences were observed between groups in lipid peroxidation (F(7,38) = 0.957, *p* = 0.476). In the analysis of the AChE enzyme, although changes were observed between groups (F(7.38) = 7.530, *p* = 0.0001), none of them involved the THY and 24-EPI groups relative to the control groups. Regarding the activity of the LDH enzyme, the only change was a decrease in its activity seen in CuSO_4_ + THY compared to the CuSO_4_ group (*p* = 0.02) (F(7,38) = 3.368 *p* = 0.007). Regarding the quantification of NO, the larvae in the CuSO_4_ group presented increased levels in relation to the E3 (*p* = 0.002), while DIC (*p* = 0.0003), CuSO_4_ + DIC (*p* = 0.006), THY (*p* = 0.02), CuSO_4_ + THY (*p* = 0.03) and CuSO_4_ + 24-EPI (*p* = 0.01) decreased in relation to the CuSO_4_ group (F(7,24) = 4.733, *p* = 0.002).

## 4. Discussion

Currently, several plant compounds need research to confirm their potential anti-inflammatory and antioxidant properties with minimal side effects [51]. This study aimed to evaluate the anti-inflammatory, anti-apoptotic and antioxidant characteristics of THY and 24-EPI, both natural compounds present in a wide variety of plants, using zebrafish larvae as an animal model. The results obtained using Tg(*mpx*GFP)^i114^ larvae showed that THY and 24-EPI have the ability to inhibit neutrophil migration to damaged tissue at the concentration of 20 µM and 5 µM, respectively. Regarding the results obtained with the AB-WT strain, both natural compounds reduced cell death without changing the majority of the analyzed oxidative stress parameters. In addition, they proved to be efficient in protecting against the action of the proinflammatory chemical agent (CuSO_4_ 10 µM). 

The transgenic zebrafish neutrophil migration model represents a promising target to study the therapeutic properties of several substances because it offers data on the inflammatory response in vivo and in a short time [52]. In this model, neutrophil migration increases considerably 3 h after injury, reaching its peak after 4 h and decreasing thereafter [33,53,54,55]. Thus, we chose to assess the anti-inflammatory profile of the compounds through the neutrophil count after 4 h. It should be noted that the mechanisms involved in neutrophils migration after injury are complex [56,57,58]. As comparison, we used compounds with recognized proinflammatory and anti-inflammatory action, CuSO_4_ 10 µM and DIC 1.5 µM, respectively [25,26]. The exposure to copper resulted, as expected, in an increase in the number of neutrophils migrating to the injured area and the exposure to DIC showed predictable results by inhibiting their migration [26]. The DIC, as NSAID, has its main mechanism of action by inhibiting COX activity [59]. However, the mechanism involved in the zebrafish larvae neutrophil migration modulation seems dependent of prostaglandin E2 for some authors [60], but its inhibiting appears to be independent of COX inhibition and prostaglandin E2 release for others [61]. Moreover, the ROS signaling may be involved in this process [56]. Nagoor Meeran et al. (2017) [8] have reviewed the anti-inflammatory properties of THY and showed its involvement in several pathways although it could be used in a similar fashion to NSAIDs. Our results showed that THY also has the ability to inhibit neutrophil migration following an inflammatory process. Previous works showed that thyme oil, where THY is the main substance, suppressed ovine neutrophil’s anti-inflammatory function following its exposition [62], as well as decreased the number of neutrophils following tail injury in zebrafish embryo [63]. The 24-EPI was also able to inhibit neutrophil migration, and our work shows, for the first time, that it can have anti-inflammatory properties. The brassinosteroids, as steroid hormones with close similarities to cholesterol-derived animal steroids have shown anti-inflammatory roles in several situations [3,10]. Previous 24-EPI testing also described its antioxidant, anti-apoptotic, and cell protection properties [10,13], which are aspects with preponderance in neutrophil migration and inflammatory processes, apart from possible specific mechanisms to be unraveled in the future.

Inflammation can be developed as one of the biochemical responses to the deleterious effects of oxidative stress, that occurs due to the imbalance of ROS production and detoxification in the organism [64]. ROS and organic peroxides are generated by mitochondrial oxidative metabolism during cellular respiration, and, in hypoxia conditions, NO can also be produced [65]. Overproduction of these molecules, especially over a prolonged period, can cause damage to cell structures and functions, resulting in cell death by necrotic and apoptotic processes [66]. Our results show that CuSO_4_ increased cell death evaluated by AO staining. This is in agreement with previous studies showing the induction of copper toxicity in fish at similar developmental stages with the induction of cell death [31,67]. However, in our case, ROS levels were not clearly increased with exposure to CuSO_4_, contrary to what was described in the aforementioned studies, in which the induction of cell death resulted from increased oxidative stress. The natural compounds analyzed showed the ability to reduce the levels of cell death, especially in those with the addition of CuSO_4_. The treatment with THY showed cell protection capacity through apoptosis reduction, and antioxidant properties by reducing ROS production. It has already been reported that THY has the ability to release its hydrogen proton from the OH group and thus inhibit the initiation of chain reactions mediated by harmful free radical molecules [68,69]. In fact, its anti-apoptotic ability has been associated with its antioxidant properties [70]. The 24-EPI also prevents cell death as demonstrated in neuronal cell culture [13]. In this sense, the 24-EPI exposure to neurotoxic-deficient zebrafish larva model allowed its recovery associated with the normalization of oxidative stress levels [10,15]. Larvae exposed to the DIC also showed low levels of ROS, and additionally, the ability to increase the levels of the antioxidant enzyme SOD. This enzyme is the first line of antioxidant defense and is the main enzyme responsible for compensating oxidative effects [13,71]. The low levels of ROS may be justified by the compensatory mechanism of SOD activity, since DIC has already been shown to increase SOD levels [72]. The CuSO_4_ + 24-EPI exposure also showed high levels of SOD, not observed in single agent exposure to 24-EPI. Indeed, an increase in SOD and other enzymes of the first line of antioxidant defense (CAT and GPx) would be expected on the basis of 24-EPI exposure in neural cells [13], but no changes in these parameters were observed in previous zebrafish studies [10,15]. On the other hand, brassinosteroids have shown the ability to limit the increase in metals in plant cells [73,74], as well as to activate the formation of specific ligands such as phytochelatins, which, by binding to metal ions, increase the capacity of antioxidant enzymes to decontaminate ROS production induced by the accumulation of heavy metals [74]. Yet, none of the treatments analyzed in the remaining biochemical parameters of antioxidant enzymes, such as CAT, GST, GPx, GSH, and GSSG, showed changes compared to the control, the same occurring in lipid peroxidation (TBARS) and energy metabolism (LDH) parameters. Based on these data, we can see some limitations of our positive control (CuSO_4_) in terms of the change of these parameters. The possible justification for this fact has to do with the insufficient exposure time [24]. Thus, the antioxidant capacity of THY and 24-EPI was limited at this level. Still, the absence of alterations induced by THY alone confirms that the concentration used does not present a pro-oxidant effect [6]. Regarding the assessment of changes induced by neurotransmission, treatment with CuSO_4_ + DIC impacted the cholinergic system promoting a reduction in the enzymatic activity of AChE, although the same results were not observed when tested in single exposure. This may have been due to the combination of compounds since another study demonstrated that the association between copper and another NSAID (sodium meclofenamate) was able to induce an inhibitory effect on this enzyme [75]. CuSO4 behaved as expected, inducing an increase in the molecular production of NO, which can be corroborated by previous work where zebrafish larvae were exposed to the same concentration [24]. NO is an inorganic free radical involved in the pathological process of inflammation and is also associated with promoting apoptosis in some cells [76,77]. The increase in nitric oxide in inflammatory processes in which neutrophils are increased is widely described [78], consequently the reduction in the number of neutrophils that migrated to the site of injury in exposure to THY and 24-EPI may explain the lower levels of NO in these treatments. Natural compounds exert their anti-inflammatory activity by blocking NF-*κ*B and mitogen-activated protein kinase (MAPK)-dependent signaling pathways [4]. These are involved in the production of several pro-inflammatory mediators, including iNOS, which have been described to be blocked by thymol [79]. So far, we do not know the direct effect of 24-EPI on NO production. Notwithstanding, it has been found to inhibit glycogen synthase kinase 3β in vitro models and Caenorhabditis elegans [80], which indirectly facilitates interferon-γ-induced nuclear factor-κB activation and nitric oxide biosynthesis [81]. Another possibility, which requires future investigation, is that THY and 24-EPI, being heterocyclic compounds, may have the ability to inhibit nitric oxide synthase 2b, as reported for the heterocyclic compound 7-Nitroindazole, resulting in a reduction in neutrophil recruitment in a CuSO4 induced zebrafish model of inflammation [82].

## 5. Conclusions

In summary, we demonstrate that the phenolic compound THY and the phytosteroid 24-EPI showed the ability to decrease neutrophil migration in Tg(*mpx*GFP)^i114^ larvae. These compounds also have anti-inflammatory properties capable of preventing oxidative stress and protection from cell death of WT larvae. Furthermore, these natural products demonstrated a great ability to inhibit copper-induced oxidative, NO increase and apoptotic processes. The results derived from this study may direct future works toward the exploration and identification of the anti-inflammatory mechanisms of action of these compounds. 

## Figures and Tables

**Figure 1 antioxidants-12-01297-f001:**
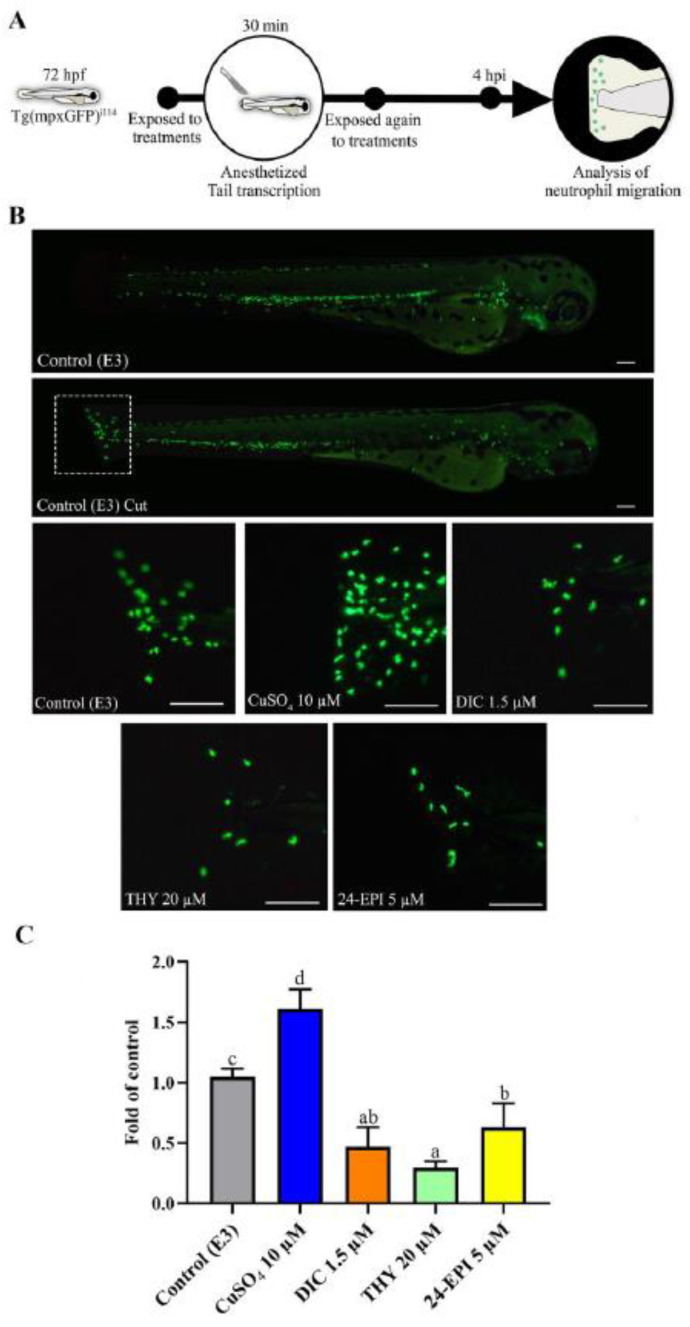
Treatment effects on neutrophil migration to the lesion site in zebrafish larvae (Tg(*mpx*GFP)^i114^) produced by tail transection. (**A**) Diagram of the experimental design for the neutrophil migration (**B**) Image of a normal Tg(*mpx*GFP)^i114^ zebrafish larva (control), and a larva with tail transection (Control cut) and detailed photos of the transection site after 4 h of treatment. The scale bar represents 125 μm. (**C**) Graph showing the normalized number of neutrophils that migrated to the tail after 4 hpi (Mean of control: 30.7 ± 9.2). Data are expressed as mean ± SD from at least five independent samples from five random animals each. Different letters represent statistical differences among treatment groups (*p* < 0.05).

**Figure 2 antioxidants-12-01297-f002:**
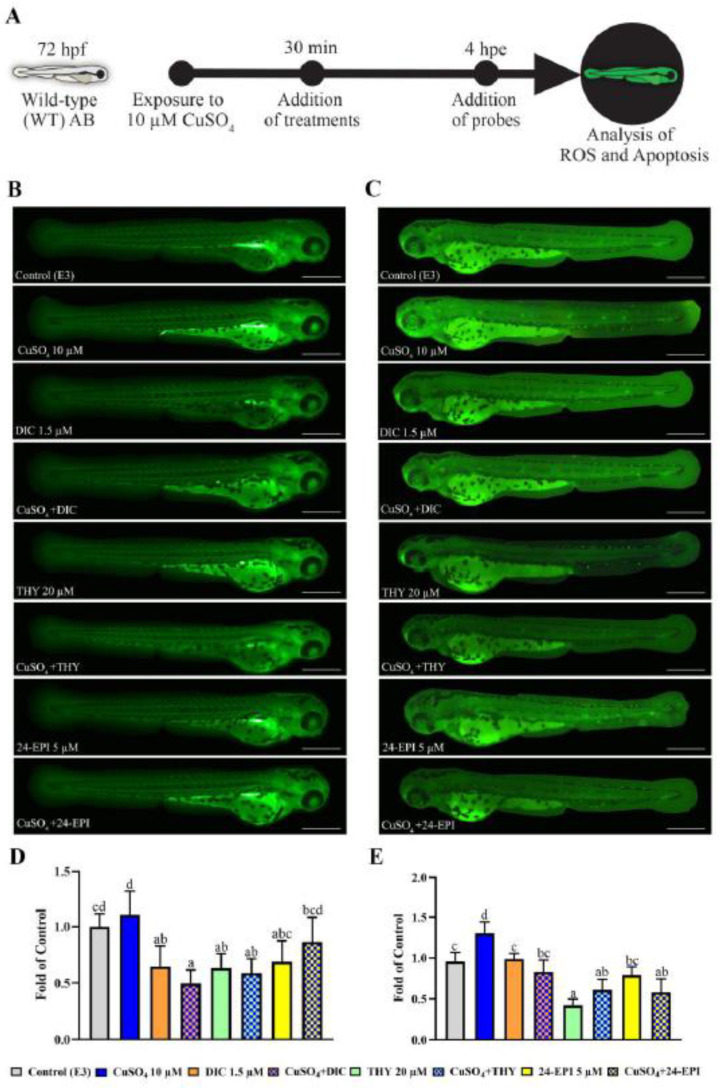
Effect of different treatments on cell death and ROS production in zebrafish WT larvae with 72 hpf exposed for 4 h. Data from at least five independent samples from twenty-five random animals each. (**A**) Schematic diagram showing the experimental exposure protocol for the study of cell death and ROS production. (**B**) Illustrative images from larvae exposed to the DCF probe. (**C**) Illustrative images from larvae exposed to the AO probe. (**D**) Result of the DCF fluorescence intensities in homogenized larvae (Mean of control: 37.9 ± 8.8). (**E**) Result of AO fluorescence intensities in homogenized larvae (Mean of control: 48.4 ± 7.5). Data are expressed as mean ± SD and normalized according to the control group. Different letters represent statistical differences among treatment groups (*p* < 0.05).

**Figure 3 antioxidants-12-01297-f003:**
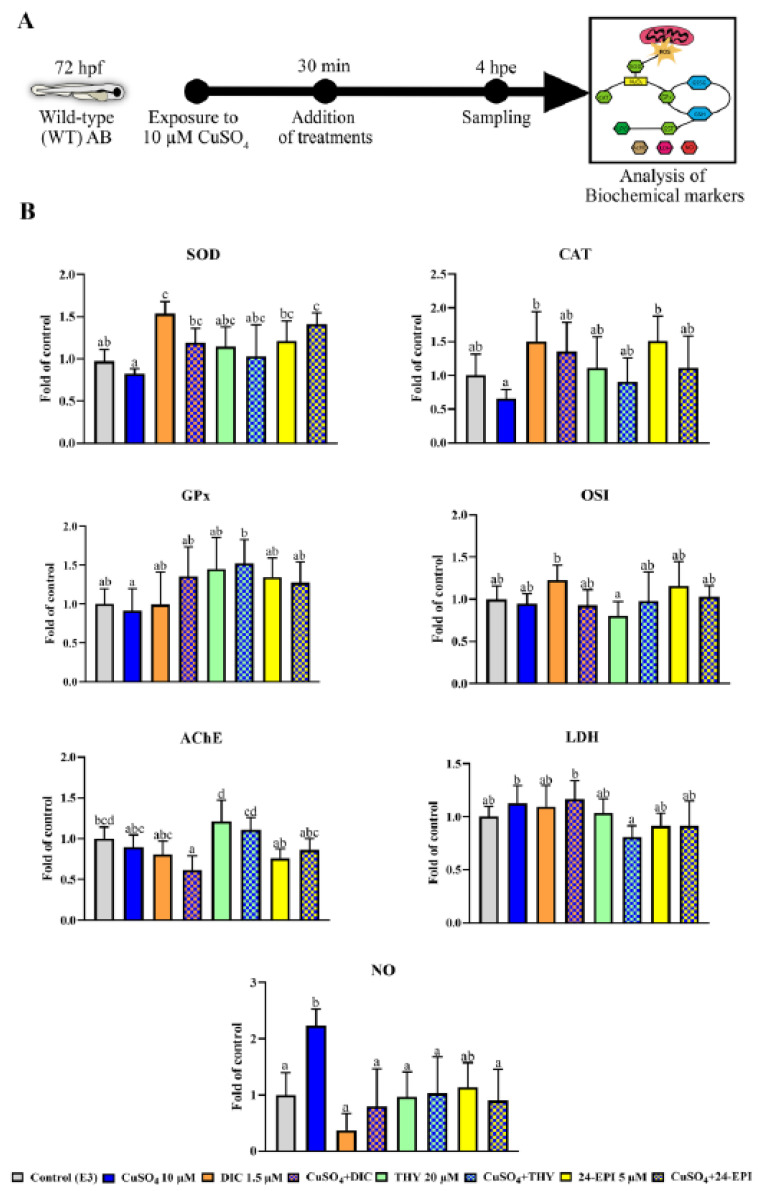
Biochemical indicators were examined in 72-hour-old zebrafish WT larvae subjected to various treatments for 4 h. (**A**) Schematic diagram of the experimental methodology for biochemical marker exposure and analysis. (**B**) Graphs of biochemical indicators that differed significantly following exposure to various treatments. Data were obtained from at least five independent samples from fifty random animals each and values normalized to the control group. Data are expressed as median (interquartile range) for non-parametric data (Median of control: SOD = 7.2 (6.3–7.9) U/mg.protein) or mean ± SD for parametric data distribution (Mean of control: CAT = 2.9 ± 0.9 U/mg.protein; GPx = 2.24 ± 0.3 nmol NADPH/min.mg protein; OSI = 0.18 ± 0.05; AChE = 10.43 ± 1.96 μmol TNB/min.mg protein; LDH = 65.45 ± 5.15 nmol NADH/min.mg protein; NO = 23.4 ± 11.3 nmol NO/mg protein). Different letters represent statistical differences among treatment groups (*p* < 0.05).

## Data Availability

The data that support the findings of this study are available from the corresponding author upon reasonable request.

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
