# Peer review of "Anti-Inflammatory, Anti-Oxidative and Anti-Apoptotic Effects of Thymol and 24-Epibrassinolide in Zebrafish Larvae"

_antioxidants, 2023, doi:10.3390/antiox12061297_

Round 1

Reviewer 1 Report

I have read the text very carefully and am very impressed by the good quality of the work. I could go on and on about why exactly these two substances were chosen for the study. However, in view of the excellent quality of the work, such a question would be out of place. It is immediately apparent that the team of authors knows what they are doing. I am in favour of accepting the paper in its present form (after explanations about ethics).

I just have one question about getting approval from the ethics committee. It seems that larval forms capable of feeding themselves require such approval. Four to five days after fertilisation, the zebrafish larvae are approx. 3 mm and become capable of independent food intake food. This is also the point at which the zebrafish larvae becomes an becomes an animal in accordance with the legislation and therefore only from this point onwards do procedures need to be authorised by the ethics committee. Please explain.

Author Response

  1. I just have one question about getting approval from the ethics committee. It seems that larval forms capable of feeding themselves require such approval. Four to five days after fertilisation, the zebrafish larvae are approx. 3 mm and become capable of independent food intake food. This is also the point at which the zebrafish larvae becomes an becomes an animal in accordance with the legislation and therefore only from this point onwards do procedures need to be authorised by the ethics committee. Please explain.

Response:  The authors appreciate the comment by the reviewer which has already been required by the editorial team before the manuscript went peer review. In accordance, lhe license information was included in the revised manuscript.

Reviewer 2 Report

Title: Anti-inflammatory, anti-oxidative and anti-apoptotic effects of 2 thymol and 24-epibrassinolide in zebrafish larvae.

General comments: The manuscript by Lanzarin and colleagues makes an important contribution in the study of new natural active compounds and their effect in order to be valid tools for future applications. Nevertheless,I have few concerns about the materials and methods section, specifically the part focused on the experimental design.

Line 22: write Danio rerio in italic and uniform the entire text

I suggest that the authors rework the structure of the experimental design, it is in fact confusing and it is not clear either the number of total larvae used or how many are WT or how the experimental groups are divided and how many larvae were used for each analysis. I suggest that the authors better structure this section and take a cue from this publication https://doi.org/10.3390/toxics10050272 and cite it so as to make the discussion and understanding of both the individuals used for each experimental group and the samples used for each analysis more fluid.

Title: Anti-inflammatory, anti-oxidative and anti-apoptotic effects of 2 thymol and 24-epibrassinolide in zebrafish larvae.

General comments: The manuscript by Lanzarin and colleagues makes an important contribution in the study of new natural active compounds and their effect in order to be valid tools for future applications. Nevertheless,I have few concerns about the materials and methods section, specifically the part focused on the experimental design.

Line 22: write Danio rerio in italic and uniform the entire text

I suggest that the authors rework the structure of the experimental design, it is in fact confusing and it is not clear either the number of total larvae used or how many are WT or how the experimental groups are divided and how many larvae were used for each analysis. I suggest that the authors better structure this section and take a cue from this publication https://doi.org/10.3390/toxics10050272 and cite it so as to make the discussion and understanding of both the individuals used for each experimental group and the samples used for each analysis more fluid.

Author Response

  1. Line 22: write Danio rerio in italic and uniform the entire text.

Response:  Changes were made accordingly, and the entire manuscript revised.

  1. I suggest that the authors rework the structure of the experimental design, it is in fact confusing and it is not clear either the number of total larvae used or how many are WT or how the experimental groups are divided and how many larvae were used for each analysis. I suggest that the authors better structure this section and take a cue from this publication https://doi.org/10.3390/toxics10050272 and cite it so as to make the discussion and understanding of both the individuals used for each experimental group and the samples used for each analysis more fluid.

Response: The authors appreciate the comment raised by the reviewer. For a better understanding, we have reviewed the experimental design, the number of larvae and replicates used in each phase of the study, which is also indicated in the respective figures of each phase of the study.